# The Expansion of the Spectrum in Stuttering Disorders to a Novel ARMC Gene Family (*ARMC3*)

**DOI:** 10.3390/genes13122299

**Published:** 2022-12-06

**Authors:** Adil U Rehman, Malaika Hamid, Sher Alam Khan, Muhammad Eisa, Wasim Ullah, Zia Ur Rehman, Muzammil Ahmad Khan, Sulman Basit, Noor Muhammad, Saadullah Khan, Naveed Wasif

**Affiliations:** 1Department of Biotechnology and Genetic Engineering, Kohat University of Science and Technology (KUST), Kohat 26000, KP, Pakistan; 2Gomal Centre of Biochemistry and Biotechnology, Gomal University, Dera Ismail Khan 29111, KP, Pakistan; 3Center for Genetics and Inherited Diseases, Taibah University, Al-Madinah Al-Munwarah 42238, Saudi Arabia; 4Institute of Human Genetics, Ulm University and Ulm University Medical Center, 89081 Ulm, Germany; 5Institute of Human Genetics, University Hospital Schleswig-Holstein, Campus Kiel, 24106 Kiel, Germany

**Keywords:** autosomal recessive, stuttering, exome sequencing, splice site variant, *ARMC3*

## Abstract

Stuttering is a common neurodevelopment speech disorder that negatively affects the socio-psychological dimensions of people with disability. It displays many attributes of a complex genetic trait, and a few genetic loci have been identified through linkage studies. Stuttering is highly variable regarding its phenotypes and molecular etiology. However, all stutters have some common features, including blocks in speech, prolongation, and repetition of sounds, syllables, and words. The involuntary actions associated with stuttering often involve increased eye blinking, tremors of the lips or jaws, head jerks, clenched fists, perspiration, and cardiovascular changes. In the present study, we recruited a consanguineous Pakistani family showing an autosomal recessive mode of inheritance. The exome sequencing identified a homozygous splice site variant in *ARMC3* (Armadillo Repeat Containing 3) in a consanguineous Pashtun family of Pakistani origin as the underlying genetic cause of non-syndromic stuttering. The homozygous splice site variant (NM_173081.5:c.916 + 1G > A) segregated with the stuttering phenotype in this family. The splice change leading to the skipping of exon-8 is a loss of function (LoF) variant, which is predicted to undergo NMD (Nonsense mediated decay). Here, we report *ARMC3* as a novel candidate gene causing the stuttering phenotype. *ARMC3* may lead to neurodevelopmental disorders, including stuttering in humans.

## 1. Introduction

Stuttering is a genetic form of a speech disorder [1], characterized by involuntary interruptions of verbal fluency, audible or silent repetitions, prolongations, and blockage in sounds, words, or syllables during a conversation. It is a non-controllable condition associated with other phenotypes, such as movements and emotions of adverse effects, including fear, embarrassment, or irritation [2,3]. Repetition is the primary concern of many individuals who stutter. This abnormality usually occurs in children at the time when they are in their developing stage of language abilities. Stuttering has a more prevalent effect on males, with a 3:1 ratio versus females. It occurs in approximately 5–8% of children between two and four years of age [4,5].

Stuttering may be classified into two types: neurogenic/acquired and developmental stuttering. In the former case, the causative agents may include stroke, head injury, brain tumor, neurological impairment, or aftershocks of treatment for neurological diseases [6,7,8]. Developmental stuttering is more common and affects 3–8% of preschool-aged children at two to five years of age; however, it resolves spontaneously in 75% of cases [2,9]. Developmental or persistent developmental stuttering (PDS) is a fluency disorder that appears during early childhood without any apparent physical causes. PDS is generally used for cases that are not spontaneously resolved. It persists in ~1% of the adult population and occurs in all cultures and ethnicities. Based on the severity of the condition, PDS seriously affects the quality of life [2,10].

So far, studies concerning stuttering are involved; it displays a highly heritable mode of inheritance [11]. In most cases, it shows an autosomal recessive mode of inheritance [12]. Through previous linkage studies, four loci have been proposed for PDS, including STUT1 (OMIM: 184450), STUT2 (OMIM: 609261), STUT3 (OMIM: 614655), and STUT4 (OMIM: 614668). STUT1 locus is located on chromosome 15q21.1 and was identified in the genome-wide linkage studies of North American and European families [13]. In several studies on Pakistani stuttering families, other novel PDS loci were identified, including STUT2 on chromosome 12 [14], STUT3 on chromosome 3q13.2-q13.33 [15], and STUT4 on chromosome 16q [16]. Previously, pathogenic variants in various genes, including *GNPTAB* (OMIM: 607840; 12q23.2), *GNPTG* (OMIM: 607838; 16p13.3), *NAGPA* (OMIM: 607985; 16p13.3), *AP4E1* (OMIM: 607244; 15q21.2), and *IFNAR1* (OMIM: 107450; 21q22.11) [13,17,18] underlying stuttering phenotypes have been identified using linkage studies and exome and Sanger sequencing.

With the recent advanced techniques of next-generation sequencing (NGS), the decoding of DNA has been automated, and new genes/variants are identified, causing disease phenotypes in the affected individuals. Here, we present phenotypic manifestations and genetic analyses of a consanguineous Pashtun family of Pakistani origin, segregating non-syndromic PDS in an autosomal recessive manner. Furthermore, the exome sequencing data analysis revealed a splice donor site variation (NM_173081.5; c.916 + 1G > A) in *ARMC3* that most likely causes the non-syndromic persistent developmental stuttering.

## 2. Material and Methods

### 2.1. Ethical Approval, Subject Recruitment, and Blood Collection

The Advanced Studies and Research Board (ASRB) and the Research and Ethical Committee of Kohat University of Science and Technology, Kohat, Khyber Pakhtunkhwa (KP), Pakistan, approved this study.

In this study, a five-generation consanguineous Pashtun family, with two affected individuals, V-2 and V-4 (V-4 died one year after the blood collection), with all hallmarks of PDS, was recruited from the remote southern region of Khyber Pakhtunkhwa, Pakistan (Figure 1). First, the elderly family members were interviewed about the family history of the disorder and its phenotypes. Then, the pedigree was constructed based on the provided information. The pedigree convincingly showed an autosomal recessive mode of inheritance. After that, 5–6 mL of venous blood was collected from six participating members (IV-3, IV-4, V-1, V-2, V-3, and V-4) upon signing the written informed consent form.

### 2.2. Genomic DNA Extraction

Genomic DNA (gDNA) was extracted from whole blood samples using the QIAamp DNA Mini Kit (Qiagen, Hilden, Germany). The purity of the DNA was measured by Qubit^®^ 2.0 Fluorometer (Invitrogen, Carlsbad, CA, USA) and NanoDrop ND-1000 (Thermo Scientific, Wilmington, DE, USA), respectively.

### 2.3. Exome Sequencing

A total of 80 ng/µL of the extracted gDNA of the proband (V-2) was used for the exome sequencing. The sequencing was performed by using the NimbleGen SeqCap EZ Human Exome Library v3.0 (Nimblegen 64 Mb) (Roche NimbleGen, Madison, WI, USA) following the manufacturer’s protocols [19]. The enriched libraries were sequenced by the Illumina Hiseq2000 platform (Illumina, Inc., San Diego, CA, USA) with 90-bp paired-end reads. Using the Burrows–Wheeler Aligner (BWA) and SOAPaligner with default parameters, the reads data were mapped to human reference genome 19 (NCBI build 37.1, hg19) assembly. SOAPsnp (soap2.21) and GATK (version v1.0.4705) were used for annotating the single-nucleotide variants (SNVs) and insertion–deletion (INDELS) variants [20,21]. Low-quality data (cover depth < 20) and high-frequency variants, having a minor allele frequency (MAF) of >0.001 in three public databases, including ESP500 (https://esp.gs.washington.edu/drupal/ accessed on 10 June 2021), 1000 Genome Project (http://www.1000genomes.org/ accessed on 10 June 2021), and gnomAD (http://exac.broadinstitute.org/ accessed on 10 June 2021), in the general population were filtered out from the raw data. After filtration, the VaRank program was used for candidate variant prioritization [22]. Several huge runs of homozygosity (ROHs) were identified using the AgileVCFMapper tool. Moreover, CNMOPS and ExomeDepth algorithms were used for the coverage analysis of CNV detection. Eventually, the data and the annotation of functional variants were combined through COMBINE and FUNC algorithms.

### 2.4. Variant Search, Classification, and Sanger Sequencing

For the exome data analysis, variant calling files (VCFs) were generated via the VARBANK exome pipeline v2.26 from the Cologne Centre for Genomics (CCG, https://varbank.ccg.uni-koeln.de/ accessed on 1 July 2021). The previously reported stuttering-causing genes (*GNPTAB*, *GNPTG*, *NAGPA*, *AP4E1*, and *IFNAR1*) were filtered out to exclude the inclusion of their copy number variants, missense, nonsense, or compound heterozygous variations.

First, the runs of homozygosity (ROHs) in the affected member were identified based on the consanguinity among their parents. In the next step, a variant search was carried out in the ROHs to identify the rare homozygous variants. Furthermore, the exome-wide search was conducted to search for rare homozygous variants. Three datasets were used to evaluate the variants, including VarSome, HGMD professional 2021.4, and dbSNP. The MAF value of <0.01 for the variants was established by consulting the Genome Aggregation Database v2.1.1 (gnomAD; https://gnomad.broadinstitute.org/ accessed on 10 June 2021). An in-house database of 512 exomes of patients with diverse phenotypes and another dataset of 31 exomes of ethically matched Pashtun patients, along with 100 exomes of other Pakistani patients of various ethnic backgrounds, were used as controls. According to the ACMG guidelines, variables were classified as pathogenic with uncertain significance and pathogenic for ARMC3 and BCHE variants, respectively [23]. For the *CACNA1F*, no such prediction was found.

Moreover, considering the gender of the affected members (V-2, V-4) in the pedigree chart, a thorough search was conducted on the variant data of the X-chromosome.

After executing the process of exome data analysis, various databases, including the dbSNP (https://www.ncbi.nlm.nih.gov/snp/ accessed on 10 June 2021), 1000 genome project (https://www.internationalgenome.org/ accessed on 10 June 2021), EVS (https://evs.gs.washington.edu/EVS/ accessed on 10 June 2021), and gnomAD (https://gnomad.broadinstitute.org/ accessed on 10 June 2021), were consulted for filtration and validation of the identified variants.

The UCSC (University of California Santa Cruz) genome browser (http://genome.ucsc.edu/cgi-bin/hgGateway accessed on 5 January 2022) was consulted to obtain the reference sequence. Primer 3 (version 0.4.0) (https://bioinfo.ut.ee/primer3-0.4.0/ accessed on 7 January 2022), an online tool, was used for designing the primers for amplifying the regions of interest. After PCR amplifying the regions of interest, an Exo-Sap protocol (https://www.thermofisher.com accessed on 13 February 2022) was used to purify the PCR products. The ABI3730 genetic analyzer with BigDye chemistry v3.1 was used for DNA sequencing. For alignment of the query sequences with the reference sequence, a sequence alignment editor, BioEdit version 6.0.7 (http://www.mbio.ncsu.edu/BioEdit/bioedit.htm, accessed on 19 March 2022), was used.

### 2.5. Pathogenicity Prediction

The pathogenicity of *ARMC3* splice site variant c.916 + 1G > A was determined through MutationTaster, VarSome, CADD, VarSEAK, RegSNP, Human Splice Finder (HSF), MaxEntScan, FATHMM-MKL, EIGEN, EIGEN PC, BayesDel noAF, PhyloP100way, and GERP tools. In addition to these, PolyPhen-2, SIFT, and PROVEAN were also used for the missense variants of *BCHE* and *CACNA1F*.

### 2.6. Protein Structure Modeling and Interaction Studies

The mutant protein was predicted by skipping the exon-8 of the *ARMC3* transcript (NM_173081.5) and then translated using the ExPASy translation tool.

A Protein 3D model was predicted through an online tool, “I-TASSER” [24]. The predicted structure of normal and mutated ARMC3 protein was then docked with its close functional interactor “MYCBPAP” using ClusPro software (https://cluspro.org accessed on 12 May 2022) [25]. The protein interactor was identified through the STRING database (string-db.org accessed on 13 September 2022), a functional protein association network [26]. The predicted structure and interactions were visualized through LigPlot+ (Version 2.1) [27] and Chimera 1.13.1 (https://www.cgl.ucsf.edu/chimera/ accessed on 13 May 2022) [28].

## 3. Results

### 3.1. Clinical Phenotypes of the Patients

According to the parents, both the affected individuals (V-2 and V-4) were born through expected delivery and had never faced any accidental emotional trauma or brain damage. There was no history of stuttering in this family. By interviewing their parents, we found that both patients suffered from persistent developmental non-syndromic stuttering in their early childhood. The phenotypes were confirmed by asking a series of questions from both patients (V-2, V-4). Patient V-2 was a 19-year-old, mild stutterer with repetition of words but sometimes blocks at the beginning or the middle of the sentence during a conversation. He showed some secondary features in the form of involuntary actions associated with stuttering, such as blinking of eyes, perspiration, facial changes, and movements of hands and head during speech (Table 1).

Patient V-4 suffered from severe PDS (non-syndromic), and blocks could be observed at the beginning of the sentence. Sometimes, he gave up his attempt during a conversation. He used to avoid talking and preferred writing on paper or a whiteboard during school. He also exercised the involuntary closing of eyes, body movements, and facial muscle tension. He had a repeated history of nervousness during his conversation, especially with strangers, teachers, and elders (Table 1). The problem mostly worsened during fear, fatigue, or when asked questions and excitement. His social interaction was negligible with his contemporaries.

By inquiring from their parents, it was confirmed that neither the patients had any history of developmental delay nor any signs and symptoms of facial dysmorphism, skeletal and joint anomalies, eye problems, and heart-related complications. Furthermore, their intellects were average, with no neurological abnormalities.

### 3.2. Variant Identification and Co-Segregation Analysis

Exome sequencing analysis excluded the involvement of any previously reported stuttering genes in the disease etiology; however, huge runs of homozygosity (ROHs), including 57.38 MB and 27.2 MB, were found on chromosomes 10 and 3, respectively, using the AgileVCFMapper tool. A relatively small ROH of 4.4 MB was also found on chromosome 7, but no variant of interest was identified in this region. The former two chromosomal locations (10 and 3) carried two rare homozygous variants including, c.916 + 1G > A (NM_173081.5) in *ARMC3* (OMIM: 611226; 10p12.2) and c.293A > G;p.Asp98Gly (NM_000055.4) in *BCHE* (OMIM: 177400; 3q26.1). In addition, a homozygous variant c.1555G > A;p.Gly519Ser (NM_001256790.3) was also identified in *CACNA1F* (OMIM: 300110; Xp11.23). The fine-mapped variants were sequenced through Sanger sequencing for the validation and confirmation of co-segregation. During co-segregation analysis, the variant in *ARMC3* located in the ROH of 57.38 MB on chromosome 10 segregated the family with the disease phenotype. However, *BCHE* and *CACNA1F* variants did not show co-segregation in the family. Sanger sequencing analysis showed that *ARMC3* c.916 + 1G > A (NM_173081.5) variant was homozygous (A/A) in affected individuals V-2 and V-4, heterozygous (G/A) in phenotypically unaffected members (IV-3, IV-4, V-1), and a wild-type genotype (G/G) in the normal healthy individual V-3 (Figure 1 and Figure 2C).

The total allele frequency (AF: 0.00009521) for *ARMC3* variant c.916 + 1G > A in the genomAD v2.1.1 dataset was calculated. It is reported 23 times heterozygous in various populations, including East Asian (11), South Asian (7), Latino/Admixed American (3), European (non-Finnish) (1), and in other minor populations (1), with the total number of alleles 241,560, with no report of homozygote. Moreover, 22,308 South Asian alleles with an allele frequency of 0.0003138 have been reported in this database. This monoallelic variant is registered with the IDs rs767509621 and CA5437831 in dbSNP and the ClinGen Allele Registry.

### 3.3. Pathogenicity Prediction

Various online pathogenicity prediction tools were used to classify the candidate gene variants. *ARMC3* splice site variant was declared pathogenic (PVS1, PM2_supporting, PP1) based on the standards and guidelines of ACMG/AMP to interpret sequence variants [23]. The multiple pathogenicity prediction tools determined this variant to be deleterious; for example, Varseak categorized it as a class 5 variant (with a defective splice donor site). Similarly, Varsome indicated a pathogenicity score of 5, suggesting that the variant is deleterious. A high loss of intolerance probability (pLI) (4.53 × 10^−8^) score favored the notion that it is a loss-of-function variant.

Moreover, RegSNP indicated this variant as D for disease-causing, and the Human splice finder declared this variant as “probably affecting the splicing mechanism.” Likewise, MaxEntScan predicted the variant as likely disruptive. In addition to these, we also used some other online in silico tools, including FATHMM-MKL, EIGEN, EIGEN PC, and BayesDel noAF, which declared the *ARMC3* variant as pathogenic. PhyloP100way and GERP showed the variant as highly conserved, while CADD achieved the highest score of 28.8, signifying it as pathogenic (Appendix A).

### 3.4. Protein Structural Findings

The tertiary structure of both wild-type and mutant proteins showed significant changes in their folding patterns. Superimposition of protein 3D structures for wild-type and mutant revealed a low alignment score, which indicates a severe morphological impact of mutation that may compromise the protein function (Figure 3). The interaction findings of wild-type ARMC3 with MYCBPAP showed 15 interaction sites through 12 amino acid residues (Lys555, Trp502, Lys676, Arg675, Tyr721, Glu182, Asp524, Gln558, Glu471, Glu823, Glu763, Asn382), while the mutant protein was found to interact through 16 bonding sites via 13 amino acids residues (Asp523, Ala600, Tyr596, Arg422, His597, Lys593, Ser357, Arg398, Gly603, Asn386, Asp350, Arg360, Glu657). These findings also determined that altered folding patterns in mutant protein had deleteriously increased the number of interacting points, bonding pattern, and number and nature of interacting amino acid residues (Figure 4a,b). It was also observed that post-mutation interacting amino acid residues of ARMC3 were found to be entirely different from wild-type ARMC3 while interacting with MYCBPAP.

## 4. Discussion

The genetic/molecular studies of stuttering are very complex because of many factors, including its spontaneous recovery [10] and the likelihood of its non-genetic and heterogeneous causes, that is, the presence and absence of nonpenetrant mutations associated with stuttering in unaffected and affected individuals, respectively. Mutational studies on stuttering, involving the previously known genes have reported four missense variants (c.961A > G;p.Ser321Gly, c.1363G > T;p.Ala455Ser, c.1875C > G;p.Phe624Leu and c.3598G > A;p.Glu1200Lys) in *GNPTAB*, one duplication (c.11_19dup;p.Leu5_Arg7dup) and two missense variants (c.74C > A;p.Ala25Glu and c.688C > G;p.Leu230Val) in *GNPTG*, two missense (c.252C > G;p.His84Gln and c.982C > T;p.Arg328Cys) and a frameshift disease-causing variant (c.1538_1553del;p.Phe513Serfs*113) in *NAGPA* [17], and two missense variants (c.1549G > A;p.Val517Ile and c.2401G > A;p.Glu801Lys) in *AP4E1* [18]. Recently, Sun et al. (2021) identified two missense variants (c.1282A > C;p.Lys428Gln and c.1655T > C;p.Leu552Pro) in *IFNAR1* in three Chinese families. They also found a missense disease-causing variant c.902G > A;p.Gly301Glu and a 3bp deletion c.1002_1004 delTCC;p.Pro335del in the same gene in two sporadic cases of Chinese origin [13] (Appendix A).

This study identified a novel splice site variant of *ARMC3* in a consanguineous Pakistani family. The identified novel splice site change causes skipping of exon-8, which predictably removes a 62 amino acids long peptide, generating a truncated ARMC3. This deleted peptide segment affects three ARMs (Armadillos) repeats (ARM6, ARM7, ARM8). The in silico prediction identified significant alteration in the structure of ARMC3 protein, confirming the pathogenic effect of this novel splice site change. Due to protein truncation and loss of function, it is presumed that mRNA may undergo degradation through the NMD pathway.

*ARMC3* is an armadillo repeat-containing protein 3 (β-catenin-like protein) encoding a gene located on the short arm of chromosome 10 (p12.2) and contains 19 exons (Figure 2A,B). This gene’s most extended transcript, NM_173081.5 (3718 bp), is translated to produce an 872 amino acids long peptide (https://asia.ensembl.org/index.html, accessed on 13 May 2022). ARMC3 is a Cancer-testis 81 antigen (CT18) family member, containing 12 ARM (armadillo) domains. ARMs are 45 amino acid repeats and are involved in the transduction of the canonical Wnt signaling pathway [29,30]. ARM repeats containing proteins have many diverse and fundamental functions, including tissue maintenance, oncogenesis, signal transduction, development, cell structure, adhesion and mobility, cell migration, and proliferation. The human *ARMC3* is expressed in many tissues, including the brain, skeletal muscles, liver, spleen, thymus, lungs, kidneys, prostate, and testes. Moreover, a splicing variant of *ARMC3* (2439-bp) has been isolated from the human fetal brain, which encodes a 688 amino acid long polypeptide containing three distinct ARM domains [29].

The paralogs of *ARMC3* are involved in causing various genetic disease conditions in humans, e.g., *ARMC2* (OMIM: 618424; 6q21) variants cause severe asthenoteratozoospermia in humans and mice [31]. *ARMC4* (OMIM: 615408; 10p12.1) mutants show the phenotypes of primary ciliary dyskinesia (OMIM: 615451) and male infertility [32]. *ARMC5* (OMIM: 615649; 16p11.2), a tumor suppressor gene, is responsible for a familial form of bilateral macronodular adrenocortical hyperplasia (PBMAH), a rare cause of Cushing syndrome [33]. Similarly, *ARMC5* (OMIM: 615549; 16p11.2) variants have also been found to cause sporadic neuroendocrine tumors and multiple endocrine neoplasia type-1 (MEN1; OMIM: 131100) [34]. Going ahead in the list of ARM gene super-family, we find that *ARMC9* (OMIM: 617612; 2q37.1), encoding basal body protein, is involved in causing various disease phenotypes, including a syndromic form of intellectual disability [35], Vogt–Koyanagi–Harada disease [36], and Joubert syndrome-30 (JBTS30) in humans and ciliopathy phenotypes in zebrafish mutants [37]. JBTS30 phenotypes include developmental delay, motor and speech disability, and abnormal eye movements (OMIM: 617622). The *armc12*/*ARMC12* (6p21.31) encoding protein ARMC12 regulates spatiotemporal mitochondrial dynamics during spermiogenesis and is required for male fertility in humans and mice [38].

The Mus musculus *Armc3* NM_001081083 has been involved in Wnt signaling due to α-helical armadillo repeats [39]. The human ARMC3 has 81% protein identity compared to Mus musculus [40]. It is evidenced here that the Wnt signaling pathway is involved in the speech process because the expression of *FOXP2*, the hereditary speech and language impairment gene in humans [41], is regulated by Wnt β-catenin [42]. The *FOXP2* potentially interacts with genes involved in the Wnt pathway in terms of its positive and negative feedback regulation in the developing brain [43]. In addition, in teleosts, *FoxP2* expression is regulated by the Wnt pathway transcription factor Lef1 [44]. Similarly, *WNT5B* (OMIM: 606361; 12p13.33) gene variants cause delayed and/or slurred speech [45]. The dysfunction of Wnt signaling is actively involved in schizophrenia, bipolar disorder, and autism [46]. On the other hand, the ARMC proteins including ARMC4, ARMC5, ARMC6, ARMC7, ARMC8, ARMC10, and ARMCX3 regulate the Wnt signaling pathway leading to specific nervous system disorders [40].

The Human Protein Atlas database has shown high expression of ARMC3 in various parts of human (cerebral cortex, basal ganglia, cerebellum) and mouse brain (https://www.proteinatlas.org/ENSG00000165309-ARMC3/brain accessed on 27 July 2022). Moreover, ARMC3 has high levels of expression in the hippocampus and cerebellum (https://www.proteinatlas.org/ accessed on 27 July 2022), which are involved in emotions and motor function. A person’s emotional state is strongly associated with the severity of stuttering. In addition, stuttering affects motor functions, which are essential for fluency in speech [17].

To the best of our knowledge, our study reveals the first report of the *ARMC3* splice site variant responsible for causing the non-syndromic persistent developmental stuttering. We provide clinical and molecular findings of a consanguineous Pakistani stuttering family. WES was performed for the affected individual V-2. A homozygous splice donor site pathogenic variant c.916 + 1G > A was identified in the ARM supergene family member ARMC3, which is entirely plausible to be responsible for causing the underlying phenotypes of stuttering in the studied family. The current study rationalizes the previous gene identification studies in this area and strengthens the notion of the genetic etiology of stuttering.

## Figures and Tables

**Figure 1 genes-13-02299-f001:**
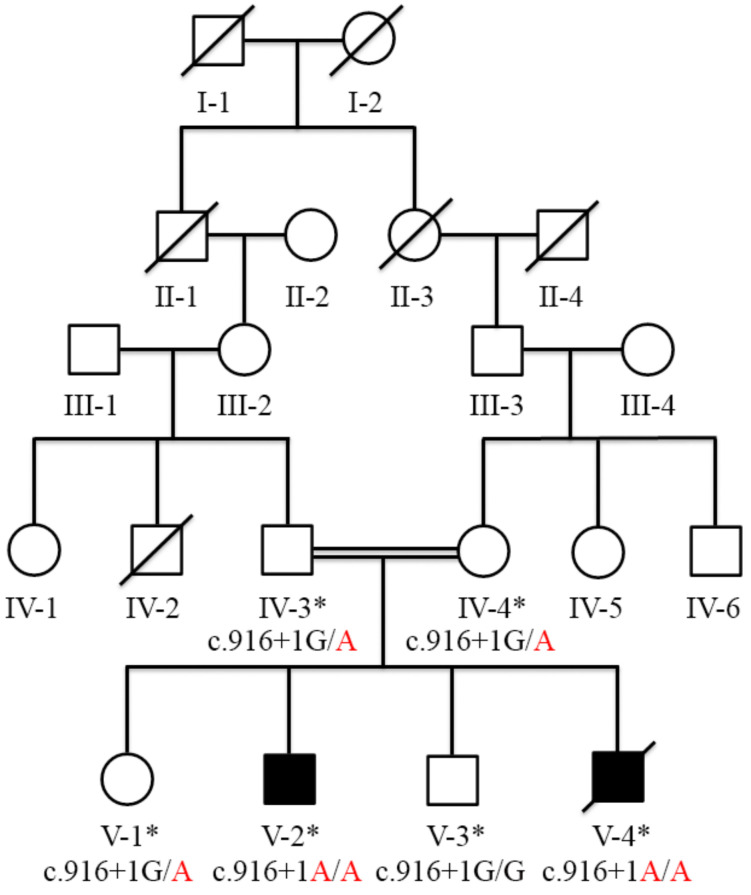
Pedigree of the family showing the segregation of the ARMC3 variant in an autosomal recessive manner. Empty squares and circles show the unaffected males and females, respectively. The filled shapes show the affected individuals. The symbols labeled with asterisks show the individuals who participated in this study. Affected members (V-2 and V-4) showed a homozygous mutant genotype (A/A). Phenotypically unaffected individuals IV-3, IV-4, and V-1 were heterozygous (G/A), while V-3 showed the wild-type genotype (G/G).

**Figure 2 genes-13-02299-f002:**
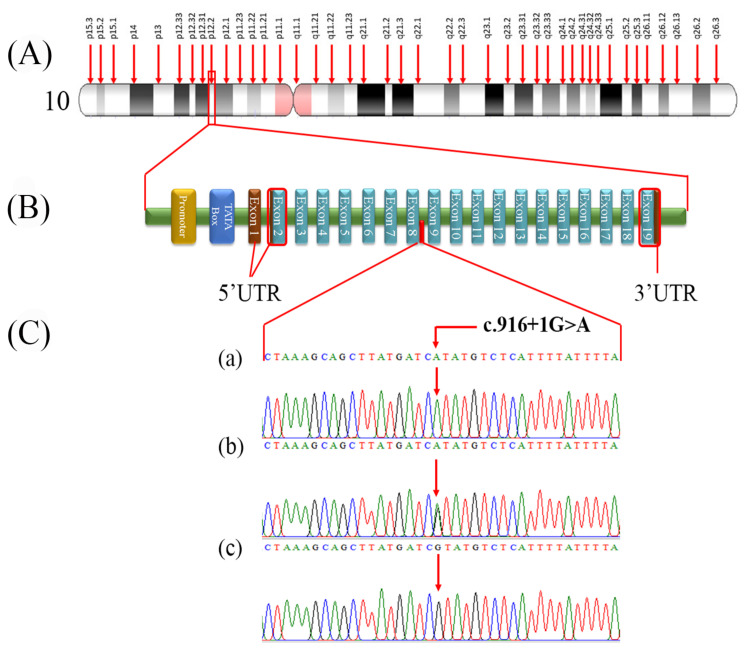
(**A**) shows the cytogenic location of the ARMC3 gene on chromosome 10; (**B**) indicates the typical structure of ARMC3 containing 19 exons and the location of the ARMC3 variant by the red bar between the exons 8 and 9. (**C**) Chromatograms. (a) Affected individual, (b) carrier, and (c) unaffected control.

**Figure 3 genes-13-02299-f003:**
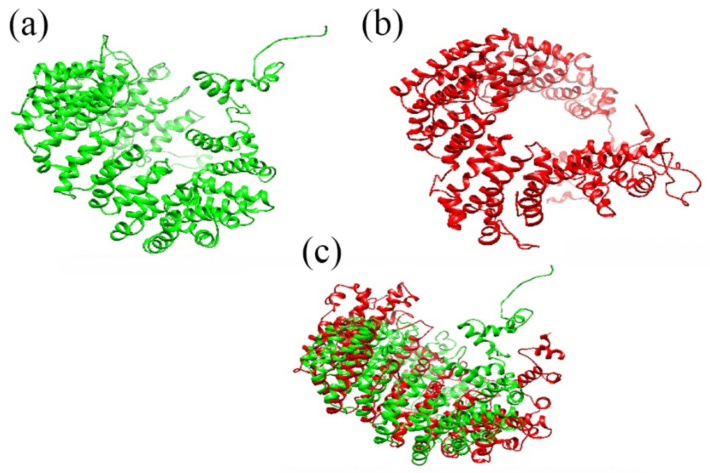
Superimposed 3D structures of wild-type and mutant ARMC3 proteins: (**a**) wild-type protein, (**b**) mutant protein, and (**c**) superimposed structure of wild-type and mutant proteins.

**Figure 4 genes-13-02299-f004:**
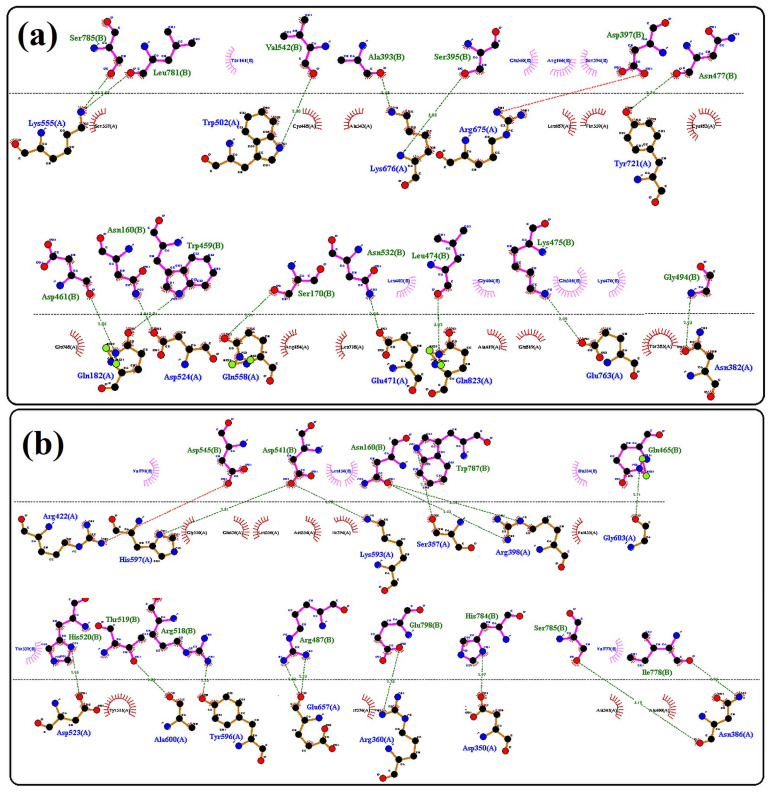
Molecular docking of the candidate proteins with the identified interactor: (**a**) docking of wild-type and (**b**) mutant ARMC3 proteins.

**Table 1 genes-13-02299-t001:** Stuttering severity assessment of the participants in this study.

Sr. No.	Participants	Syllables Stuttered	Length of Moment of Stuttering	Avoidance	Speech Tempo and Stutter Frequency	Escape
1	IV-3	NO	NO	NO	Speech tempo: High Stutter frequency: NO	NO
2	IV-4	NO	NO	NO	Speech tempo: High Stutter frequency: NO	NO
3	V-1	NO	NO	NO	Speech tempo: High Stutter frequency: NO	NO
4	V-2	Severely affected	Extremely long	Yes	Speech tempo: Low to Moderate Stutter frequency: High	Continuous extra movements of face or whole body
5	V-3	NO	NO	NO	Speech tempo: High Stutter frequency: NO	NO
6	V-4	Severely affected	Extremely long	Yes	Speech tempo: Low Stutter frequency: High	Continuous extra movements of face or whole body

NO: not observed in a particular participant.

## Data Availability

The whole exome sequencing and Sanger sequencing data is available upon resquest. ARMC3 variant is submitted to ClinVar (https://submit.ncbi.nlm.nih.gov/clinvar/) under an accession number SCV002581927 has been assigned.

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
