# Peer review of "The Expansion of the Spectrum in Stuttering Disorders to a Novel ARMC Gene Family (ARMC3)"

_genes, 2022, doi:10.3390/genes13122299_

Round 1
Reviewer 1 Report
The authors have conducted the study and presented in a manner a reader can understand its coherent methodology and discussion needs appreciation.
A speech pathologist assessment of stuttering severity and grading in the PDS affected members is important. It is also equally important to have characterized the participating members with heterozygous mutations in this case. If possible, I would advise that the authors do stuttering severity evaluation in V-2 (affected), V-1 and V-3.
Previously most of the stuttering implicated families have been largely consanguineous and attributing to single genetic etiology has been sparse and unclear. In this case, although the 5-generational family pedigree likely suggest a recessive gene involvement, it is imperative that the authors must have clarified about ruling out heterozygous mutations and its filtering strategy in variant prioritization. A flowchart depicting the filtering process could have been helpful.
The gene is not known for LOF variants (by gene constraint score) and determining the pathogenic nature of splice variant in stuttering phenotype is uncertain as discussed in this study. The authors have also carried out protein structure modeling using I-Tasser and derived interpretation based on superimposition of WT and Mutant models. However it is not clear or convincing as to how the authors are correlating the altered residues to be deleterious by means of interacting points, bonding patterns etc.,
The Post mutation interacting residues of a splice variant consequently will have different amino acids. I would also like to emphasize that the protein prediction comparison between wild type and mutant type differences required some more detailing on the loss of 44 amino acids that starts from skipping exon 9. It is also possible that the truncated protein product may undergo NMD process as the splicing may impacts more than 50% of the protein?. However, when showing protein truncation effects using protein model, it is important to detail the differences based on putative transmembrane positions.
Reviewer 2 Report
The authors identified a novel splice site variant of the ARMC3 gene in a consanguineous Pakistani family, which causes non-syndromic persistent developmental stuttering. The variant altered folding and bonding patterns of the ARMC3 protein and was predicted as pathogenic. This study highlights and disentangles genetic components of the stuttering disorder and shows decent novelties and impacts in the field of genetic aetiology of stuttering.
This reviewer thinks, in general, the manuscript is well written, and has comments to further refine the work in the following.
1. In the abstract the authors mentioned that due to complexity of the stuttering disorder, linkage studies have a little contribution to discover the genetic basis of the disorder, but in the introduction, all the previous findings on the genetic aetiology of stuttering are from linkage studies vs. population genetic studies. Such inconsistency and confusion should be avoided. Moreover, if this paper is the first NGS study on the stuttering, you should consider highlighting that, but if not, this reviewer suggests including previous NGS findings in the introduction.
2. Is there a cohort name for the Pashtun family originated from Pakistan? From the description in the material and methods, this study is cross-sectional, but if it is longitudinal, i.e., years of follow-up from the start of the cohort building, the authors should mention that.
3. In the “Variant Search, Classification, and Sanger Sequencing” section of the Methods, are the previously reported stuttering-causing genes complete? The authors mentioned 4 STUT genes in the introduction, but not excluded here.
4. What is the rationale of choosing controls? In the “Variant Search, Classification, and Sanger Sequencing” section of the Methods, the authors listed a) 512 exomes of patients with diverse phenotypes, b) 31 exomes of ethically matched Pashtun patients, and c) 100 exomes of other Pakistani patients of various ethnic backgrounds. Are there considerations and rules that selection of such controls are based on?
5. The paralogs of ARMC3 are involved in various genetic diseases related to fertility, tumorigenesis, intellectual disability, and other nervous system disorders. This review is curious about whether the identified novel splice site variant of ARMC3 is associated with speech impairment and other ARM family related disease categories in a general population. A phenome-wide association analysis, associating the novel variant of ARMC3 with different categories of diseases and relevant phenotypes in a large-scale population cohort, such as UK Biobank, Japan Biobank, Qatar Biobank etc. would be essential to answer these questions.
